# Effects of entrepreneurial orientation on social media adoption and SME performance: The moderating role of innovation capabilities

**Mingyue Fan**[1]**, Sikandar Ali Qalati**[1]*****, Muhammad Aamir Shafique Khan**[1]*****, Syed Mir Muhammad Shah**[2]**, Muhammad Ramzan**[2]**, Raza Saleem Khan**[1]

**1** School of Management, Jiangsu University, Zhenjiang, P.R. China, **2** Department of Business Administration, Sukkur IBA University, Sukkur, Pakistan

* aamirkhanju@yahoo.com (MASK); sikandarqalatiju@hotmail.com (SAQ)

**Data Availability Statement:** All relevant data are within the manuscript and its Supporting information files.

## Abstract

The increase of social media (SM) has led to continuous deviations in how day-to-day entrepreneurial activities can be carried out. Additionally, studies devoted to SM entrepreneurship and SM are relatively limited and fragmented in their focus. However there is growing interest from scholars, practitioners and academia for upcoming studies and exploration within small and medium-sized enterprises (SMEs) context. This research explores the impact of entrepreneurial orientation (EO) on SM adoption and SME performance in developing countries. We employed the resource-based view (RBV) as the foundation for developing the conceptual framework The present study employed a closed-ended questionnaire to collect data from SMEs located in Pakistan. Partial-least-squares-structural equation-modeling (PLS-SEM) was utilized for the analysis of 423 responses. The results proved a direct positive link between EO–SMEs performance, EO–SM adoption, SM adoption–SMEs performance, innovation capabilities (IC), and SME performance. Partial mediation was found between EO and SME performance, and the significant moderation effect of IC was found between SM adoption and SME performance. This paper has implications for practitioners and researchers regarding SM adoption in the SMEs. It builds an empirical, multi-dimensional hypothesized model, including mediating and moderating roles affecting the relationships.

## 1. Introduction

Gradually in today's dynamic and turbulent business environment in which organizations are challenged to face the increasing pace of innovation to improve firm performance, SM palys a fundamental role as a key stimulant to facilitate the development of entrepreneurial activities, particularly of SMEs which simultaneously contribute to economic development of the country. Since the 2008 economic crisis, the government has been given a significant importance to SMEs, as they contribute over 40% to GDP and generated over 70% of jobs. However, increasing globalization and trade liberalization have negatively affected SME performance [1].

**Funding:** "This study was supported by the National Statistical Research Project of China (2016LY96 to MF)."

**Competing interests:** The authors have declared that no competing interests exist.

Prior research suggests that, under globalization and increasing competition, SMEs having higher level of entrepreneurial orientation (EO) are likely to perform better [2]. [3] suggested that, due to limited resources and inadequate capabilities, SMEs must seek new opportunities, constantly focusing on EO. [4] posited that EO is central to strategy-making and organizational-level phenomena; therefore, to gain competitive advantage, decision-makers need to increase EO ([5]. Previous scholarships have confirmed the positive impact of EO on firm performance [6,7], but there is little understanding of this effect in emerging countries [8]. It has also been suggested that the EO–firm-performance relationship is mediated and moderated by diverse constructs [9,10], although research is limited on the causal mechanism of how and why EO impacts other constructs [11]. We aim to address this gap in an emerging country (Pakistan).

[12] proposed that to be competitive and satisfy potential customers' needs, SMEs must adopt new technologies such as social media (SM). Grounded on the RBV theory, EO is considered a key element for organizations competing in a digital environment [13]. Because entrepreneurial orientation is more supportive of adopting the new technologies and proactively responding to the changing trends, the more firm is entrepreneurial oriented, the more they will be able to compete in the industry. From an innovativeness perspective, organizations with a great extent of EO more "likely" to favor new ideas, technology adoption, and experimentation [14,15]. Also, the factor of risk-taking is involved in firms' propensity to participate in innovative projects that have indeterminate consequences [16,17]. SM, as a new interactive technology, requires managers and employees to act entrepreneurially and accept uncertain outcomes [13,18].

Continuous internet advancement has played a key role in business performance [19], and SM is particularly significant in this context in emerging countries [20]. [21] proposed that SM use in industrial markets is growing in the hope that it may foster relationships and network development, which support both profits- and innovation-related. [22] proved that SM use enables supply-chain members to improve new product development. SM-generated information also benefits industry and government business-performance policies in developing countries [23]. SM adoption has become common in business entities in developed countries and is continuing to grow in emerging countries [24,25]. SM's popularity and emerging trends facilitate firms' online learning and information-sharing processes [26], considered effective in achieving goals and improving business performance [27].

SM adoption and its effects on firm performance remain unexplored in SMEs located in Pakistan. First, empirical studies devoted to the adoption and use of SM between the firms collaborating for innovation purposes is limited [28]. Second, while few researchers proclaim that SM adoption and usage expedites innovation and entrepreneurial activities [29], others claim that empirical research on SM's benefits is insufficient [30,31]. Finally, industrial-marketing and SMEs researchers are gradually concentrating on the conditions and context under which SM affects SME performance [32,33].

According to RBV, firms' capabilities are key elements in achieving firm performance and competitive advantage [34]. [35] concluded that innovation and branding capabilities are important in this respect. This is relevant in the SM context as it is a new technology that may impact firms' innovation capabilities (IC) required to react to competitive challenges and increase firm performance. IC is considered integral to firms' strategies and a significant resource that may lead to superior performance [36]. The absence of IC negatively affects organizational knowledge acquisition and learning processes [37]. Thus, organizations' ability to innovate is vital for competitive advantage in dynamic market conditions such as SM.

Although prior work has proved significant impact of EO on firm performance, it has also been proposed that the EO–firm-performance relationship is mediated and moderated by

diverse constructs [9,10]. However, there is no consensus on the mediation effects of EO on firm performance, with limited research on the causal mechanism of how/why EO impacts other constructs such as learning, innovation, and change [11]. Recently, [38] found the positive mediation of experiential learning and networking capability between EO and business performance.

We aim to investigate the effects of EO and SM adoption on SME performance, including examining the mediation effect of SM between EO and SME performance and the moderation effect of IC. Because SMEs are a vital part of the country and their growth contributes to the overall economy and government is initiating funding programs for SMEs. This study will help SMEs to explore how to increase firm performance. In developed countries, it is already explored and studied, but there are less evidences in the context of SMEs operating in developing countries that's why this study will focus on the developing country because in Pakistan SMEs are growing, and also they are in the initial stages where they need a more clear view to proceed further as this study will help managers to make future strategies in the SMEs. This direct relationship is almost explored by researchers, but this study will explore the moderating effect of innovation capacity on SM adoption and SMEs performance. This means that this study will explore how the innovation-friendly environment in the firm makes it easy for SMEs to adopt SM quickly.

Section 2 provides a literature review covering EO–firm-performance and EO–SM-adoption relationships, the impacts of SM adoption and IC on firm performance, SM adoption's mediating role, and IC's moderating role. Section 3 details the study's methodology. The results are provided in section 4 and discussed in section 5. Theoretical and practical contributions are discussed in section 6, while section 7 examines the research limitations and future research avenues.

## 2. Literature review

### 2.1 EO and firm performance

From the RBV perspective, EO is a distinct organizational capability or intangible resource valuable in identifying, analyzing, and executing new opportunities in a way that cannot be easily substituted or imitated [39]. [40] stated that EO might serve as a source of sustainable competitive advantage and contribute to enhancing firm performance. Dynamic capability (DC) theory explains that EO can be line up with embedded higher-ordered DC, enabling firms to recognize the market opportunities, act in response to them, and reconfigure tangible capabilities to maintain competitiveness and improve firm performance [41].

EO is a strategy-making-process that escorts organizations to construct constant innovations, "adopt a proactive posture" in the industry, and commence risky-investments [41]. It attracts several activities, practices, and processes that enable organizations to act entrepreneurially. However, EO has been measured differently, e.g., using various multifaceted measures such as risk-taking, proactiveness, innovativeness, competitive aggressiveness, and autonomy [42]. Among these, proactiveness, innovativeness, and risk-taking have been widely used [5], for example, used unidimensional measures of EO based on these three dimensions.

Innovativeness relates to firms' ability to encourage and sustenance innovative ideas, experimentation, and due processes and practices that lead to new products, services, or technology [43]. Innovativeness, however, requires firms to increase investment to adopt new technology to improve IC and performance. Proactiveness reflects a firm's opportunity-seeking, forward-looking behavior, or capability to take positive measures [44]. Proactive actions lead firms to gain a competitive advantage through introducing/embracing new products or services and new technology and processes to gain a competitive advantage. At the same time, risk-taking

relates to firms' tendency to take risks for better gains [44]. Risk-taking can be viewed from various perspectives because firms engage in several types of risk-taking behaviors. From the technological ', it reveals firms' willingness to invest in technological innovations or projects with a high risk of uncertainty [45]. From a market perspective, it involves the risk of entering uncertain new markets [46]. These behavioral tendencies are commonly called EO.

EO plays a significant role in enabling organizations to achieve effectiveness; for example, EO enhances firm performance [47]. These studies indicate the considerable part of EO and that EO should be determined contextually since cultures vary across countries and within organizations. Hence, it is essential to examine the link between EO and SME performance [6,48].

Despite several studies on the EO–firm-performance link, questions remain. For example, these studies have measured EO differently based on differing EO operationalization and multi-dimensional measures: [6] measured EO by combining the work of [49] and [50]; and others have employed different EO dimension-based measures [44,51,52]. It remains still unclear which EO measures produce reliable results. The current study provides empirical evidence on EO using a more recent measure [44].

Notably, scholars have defined and measured performance differently when examining the EO–firm-performance direct link. Some have measured financial performance, while others have used subjective measures [44,51,52]. Scholars measuring SME performance have faced a similar problem [6,48]. [51] suggested that subjective performance measures will better enable decision-makers to predict the future. Therefore, we propose that the present study will provide insight to scholars and practitioners regarding the EO–SME-performance relationship by using subjective SME-performance measures. Finally, by using the time-lag approach for data collection, we answer the call for a more comprehensive method of data collection to mitigate response bias in studying the EO–SME-performance relationship. Thus, we hypothesize:

- *H1*: There is a positive relationship between EO and SME performance.

## 2.2 EO and SM adoption

Based on the RBV theory, we assume that EO is a significant factor for enterprises competing in the e-commerce business environment [53]. [18] referred to EO as "practices, methods, and decision-making styles that executives employ to act entrepreneurially." Considering the innovativeness characteristics, enterprises with higher EO are expected to be more likely to adopt new technology such as SM [44]. Particularly, EO elements such as proactivity, risk-taking, and innovativeness enable firms to adopt new or innovative technology. However, adopting innovative technologies is not risk-free; hence, it enhances uncertainty [54]. SM is a two-way communication technology, which requires managers and employees to act entrepreneurially [55] while being ready for uncertain outcomes.

However, the literature on the EO–SM-adoption link is limited. [13] stated a positive effect of SM adoption on EO in exporting firms; however, we proclaimed that an organization with great extent of EO will engage in initiatives that could bring growth, renewal, and profitability. Thus, we argue that SMEs with higher EO will adopt SM, in line with [56], who demonstrated EO's relationship with SM's perceived contribution in SMEs. However, we argue that EO's positive effects on SM adoption encompass more than specific events such as crowdfunding. Similarly, [44] examined the indirect link between EO and SM performance through small-business development and visibility; however, they did not explore any direct EO–SM-performance link. In response to the call by [20] to investigate factors that drive SM adoption in the SMEs, we hypothesize as:

- *H2*: There is a positive relationship between EO and SM adoption.

## 2.3 SM adoption and SME performance

SM is defined as internet based resource capability, a powerful enabling technology that provides synergies a complementarity with other organizations' resources [57]. [58] stated SM's importance in areas such as research and development, sales, customer support, operations, and marketing. The SM (Facebook) use by the organization has influence information accessibility, marketing information, and customer relationships [59]. SM has intense impacts on the digital world's organizations concerning handling customer interrogations, building and strengthening customer relationships, and innovative mining ideas [60]. [18] developed three sub-dimensions of SM usage (information search, marketing and branding, and building customer relationships) and found a positive relationship between them and the central construct, using reflective-formative constructs to study the different association between SM and other latent variables.

SM adoption has become common in almost every type and size of businesses [61]. Elaborating on [57] work, we use the work of [62], who stated that SM is a "group of web 2.0 based internet applications that includes blogs, forums, photo and video sharing, social networking sites, product or services reviews, online communities, etc." [63]. They enable firms to create and disseminate user-generated content. Henceforward, subsequent [64], we claim that SMEs can adopt SM to improve firm performance.

Regarding the hypothetical SM adoption–SMEs performance relationship, which is based on RBV [65], we reflect adoption of SM as a mean that assists SMEs to gain a competitive advantage [66]. This is social media platforms (i.e., Facebook) plays a critical role in easing knowledge and information sharing among external as well as internal stakeholders [67].

Third, several studies have outlined the significance of SM adoption and its usage in SMEs context [68,69]. However, the operationalization of firm performance has differed among many of these studies. Additionally, these scholarships also suggested the further examination of SM adoption and SMEs' performance. Moreover, [20] recommended that the SM adoption–SME performance should be explored in a more holistic manner employing a longitudinal approach. Therefore, we propose as following:

- *H3*: There is a positive relationship between SM adoption and SME performance.

## 2.4 IC and firm performance

IC is defined as a "firm's critical organizational capability to deploy resources in new ways to create value" [70]. This value creation helps organizations to improve their performance [71]. According to [72], IC is a means to meet the ends (enhanced SME performance); it is usually crucial for SMEs to focus on innovation as it facilitates the firms' capabilities required to respond competitively to achieve sustainable competitive advantage. According to [73], IC should be an integral part of a firm's strategy because higher innovativeness levels lead to improved cooperation and coordination within firms.

Several studies have examined the direct link between IC and firm performance [74,75]. However, these studies differed in several ways; for example, they used different conceptualizations and measurements of IC. Furthermore, [75] suggested continuous and explicit IC development at the individual and collective levels. They also recommended further investigation of this relationship in the form of surveys to improve generalizability. Following [74], who

encouraged more empirical research on IC and firm performance in other cultures, the current study investigates whether SMEs, focusing on developing countries, can cultivate IC.

Much other research [76,77]. has examined this relationship; however, the conceptualization of IC and firm performance, measurements of these constructs, the context of research, industrial sectors, study design, and respondents, have differed. They also suggest a further examination of this relationship with a longitudinal approach. Thus, the present paper aims to bring in empirical evidence employing a time lag approach to validate these empirical claims:

- *H4*: There is a positive relationship between IC and SME performance.

## 2.5 Mediating role of SM adoption

Prior research has proved the direct EO–SM relationship [20,44,56] and EO and SME performance [6,48], but very few studies have used SM as a mediator [78,79]. Prior studies suggest investigating the mechanisms that affect the direct EO–SME-performance link. For example, [38] examined the indirect effect of experiential learning on EO and SMEs' international performance. Similarly, [42] used marketing capability as a mediator to establish this link. Hence, one can infer that EO enhances SME performance based on its support from the indirect effects of other constructs.

Past studies have explained EO as a resource [38,48] that can produce valuable results (e.g., improved SME performance) only when appropriate capabilities are deployed [80,81]. RBV theory supports this, suggesting that resources should be aligned with organizational capabilities to achieve desired results. This assertion is particularly relevant to SMEs [82]. Therefore, following [17] and [45], the current study proposes SM adoption as a capability that would enable SMEs to achieve more incredible performance by capitalizing on their resources (EO in this case). We, therefore, introduce SM adoption as a mediating variable for two reasons. First, EO alone is insufficient to help SMEs improve their performance [42,83]. Thus, SMEs should borrow extra support from capabilities such as SM adoption. Second, SMEs require mechanisms to achieve what they wish to achieve [84].

We believe that SM adoption is a critical factor and it can mediate the EO–SME-performance relationship; our rationale is based on [56], who proposed that managers' perception of SM mediates the EO–crowdfunding-campaign-success link. Similarly, [20] proposed SM as a potential mechanism that indirectly affects resources and performance, while [10] suggested that the direct EO–firm-performance link is be mediated and moderated by different constructs. Recently, [69,79,85] evidenced the mediating role of SM adoption in SMEs' context in developing countries. Thus, we argue that EO enhances SME performance indirectly by affecting SM adoption:

- *H5*: SM adoption mediates the positive EO–SME performance relationship.

## 2.6 Moderating role of IC

DC theory holds that organizations produce several capabilities that enable them to structure their processes and resources to meet the organizational goals [86], allowing them to remain competitive during turbulent market conditions [87]. Nowadays, SM and innovation, though exclusively modeled, have been argued as complimentary organizational capabilities that improve firm performance. However, their complementary impact on organizational performance, is not universally linear and context-specific [88] and may have unexpected deviations from the same point, as is complexity theory's conventional argument [89]. Accordingly,

empirical investigation is required of IC's moderating conditions based on the level of EO and firm size.

Innovation has been recognized a critical factor in organizational performance as it moderates the effect of various performance antecedents [90]. However, studies have rarely used IC as a potential moderator. One rare example is [91], who incorporated IC as a potential moderator in their study on customers in a developed country.

Drawing upon RBV, we recognize SM adoption as a resource [66] that facilitates SME performance. However, to capitalize on the resource–performance link, SMEs must utilize their abilities (such as IC) to improve performance [80,91]. Our next rationale for using IC as a moderating variable is based on [20], who recommended that SM adoption–SME performance should be examined holistically. We believe that incorporating IC as an intervening variable will enable scholars and practitioners to determine whether SM adoption enhances SME performance more when ICs are effectively utilized.

Finally, IC is an SME ability that helps improve work processes and outcomes [75], hence its recognition as a potential moderator [90]. Several studies have examined the direct IC–firm-performance link [76,77]. Thus, we believe that IC strengthens the SM-adoption–SME-performance relationship:

- *H6*: IC moderates the SM-adoption–SME-performance relationship such that IC strengthens the positive SM-adoption–SME-performance relationship.

The conceptual framework is depicted in Fig 1.

## 3. Methodology

### 3.1 Ethical Statement

This study was carried out in accordance with the recommendations of the Ethical Principles of Psychologists and Code of Conduct by the American Psychological Association's (APA). All participants gave written informed consent in accordance with the Declaration of Helsinki. The protocol was approved by the employee's council of the participating organizations as well as the ethics committee of Jiangsu University, Zhenjiang, China.

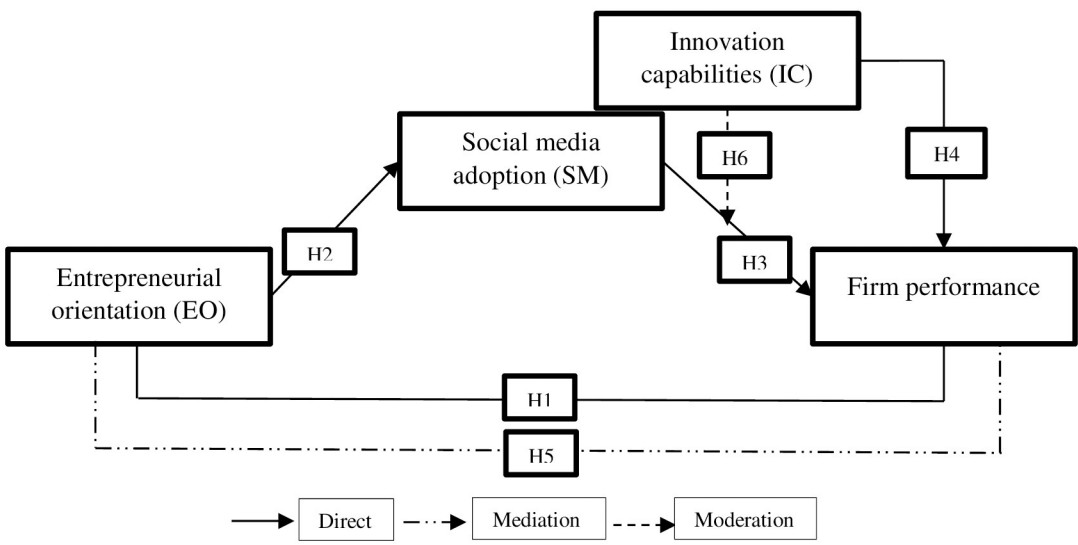

**Fig 1. Conceptual framework.**

## 3.2 Sampling and data collection

The study's targeted populations were owners and managers of the SMEs listed by small-and-medium-enterprises-development-authority (SMEDA); their online existence was confirmed using the well-known websites (i.e., businesslist.pk and mustakbil.com). We have targeted SMEs owners and managers because they are well informed about firms' performance and internal as well as external environment [92]. Besides, they make a decision regarding the adoption and implementation of technology.

In this study, a closed-ended questionnaire was administered using an online survey, which was created and designed using both survey monkey and Google Forms. Both are flexible and powerful online survey websites; they have a professional and simple layout that offers participants many functions and simple navigation of the created survey. Several other notable benefits made an online survey a more viable and appropriate source to collect data. For instance, it is an efficient way to reach many participants. It has low administration and returning costs and offers a wide range of visual elements and interactive features to attract respondents [79]. The present study's data collection process was completed in 6-month lag time as it was considered one of the ways to mitigate common method bias (CMB) [93,94]. This study distributed 650 questionnaires, resulting in 423 valid responses, representing a 65% response rate.

## 3.3 Measures

The study used five-point Likert scales (1 = "strongly disagree"; 5 = "strongly agree"). EO was assessed using eight items from [44]. SM was assessed using 13 items [SM for marketing (four items), customer relations and services (six items), and information accessibility (three items)] adapted from [18]. IC was assessed using five items from [87]. SME performance was assessed using seven items from [13,95].

## 4. Results

This study used PLS-SEM to conduct analysis [96]; as it is widely used in business management and related disciplines and is considered the most comprehensive and fully developed system of variance [97]. We also used mediation and moderation analysis [98]. In addition, the present scholarship employed two-step approach (an evaluation of measurements and structural model) as suggested by [99].

## 4.1 Descriptive information

Out of 423 respondents, 328 were male, and the remaining 22.4% were female. Nearly 38% were aged between 26 and 35 years, and almost one-fourth of them were between 36 and 45 years. Regarding education, approximately 44% had a master's degree, followed by undergraduates with nearly 30%. The most of participants were managers 40.4%, followed by executives one-third and owners one-quarter. Regarding company size, 231 had 11 to 50 employees (Table 1).

## 4.2 Model analysis

Following [99] proposition, this study used a two-step approach for PLS-SEM, namely assessment of the measurement model and the structural model. An evaluation of the measurement model includes reliability, validity, and CMB test, while the structural model comprises path coefficients, hypotheses testing, coefficient of determination, cross-validated redundancy etc.

**4.2.1 CMB test.** To ensure data free from CMB, the present study deployed Harman's singly factor test, which showed that a single factor explained only 36.44% of the total variance

**Table 1. Demographic information.**

| *Respondents* | | Frequency | Percentage |
|---|---|---|---|
| Gender | Male | 328 | 77.5 |
| | Female | 95 | 22.5 |
| Age (years) | <26 | 52 | 12.3 |
| | 26–35 | 161 | 38.1 |
| | 36–45 | 95 | 22.5 |
| | Over 50 | 115 | 27.2 |
| Education | Basic/secondary | 19 | 4.5 |
| | Undergraduate | 126 | 29.8 |
| | Master's | 184 | 43.5 |
| | Other | 94 | 22.2 |
| Position | Owner | 107 | 25.3 |
| | Executive | 145 | 34.3 |
| | Manager | 171 | 40.4 |
| No. of employees | <10 | 84 | 19.9 |
| | 11–50 | 231 | 54.6 |
| | 51–250 | 108 | 25.5 |

below the 50.0% acceptable threshold [100]. Aside from Harman's single factor test, the present study also used a full collinearity approach called the variance inflation factor (VIF) to detect CMB evidence [101]. The findings regarding CMB with this approach also ensured that data is free from CMB since the VIFs were less than 3 acceptable thresholds [102] (Table 2). [103] suggested the correlation-matrix procedure to detect CMB; whereby CMB is evident if correlation among the principle constructs is greater than 0.9 [104]; however, none of the constructs was greater than 0.9 (Table 3). Finally, following [105], in studies examining mediation effects, it is extremely difficult for respondents to manipulate mentally. Therefore, the potential for CMB is low.

**4.2.2 Assessment of the measurement model.** This research used reflective-formative approach for SM adoption construct. Therefore, we have assessed the proposed model using the first-order and second-order. [106], suggested assessing the measurement model using individual item reliability (factor loadings), internal consistency, and validity (i.e., content, convergent, and discriminant). Table 4 shows the constructs, item code, and descriptive statistics.

Individual item reliability was measured by factor loading associated with a particular dimension [107]. [108] suggested that factor loading value should be between retained 0.40 and 0.70, whereas [109] proposed a value should be ≥0.7; therefore, CR2 (0.695) was removed, while all other values were retained (Table 2). According to [110], proposition Cronbach's alpha values should be greater 0.7 (Table 2). Regarding the internal consistency [111] stated that the requires composite reliability (CR) value should be ≥0.7; all the values were retained (Table 2). Releated to convergent validity, [112] proposed that the average variance extracted (AVE) requires to be ≥0.5 (Table 2). Releated to discriminant validity, [112], stated that the AVE's square root for each variable needs to be greater than the inter-correlations of the variables with other model variables (Table 3).

**4.2.3 Assessment of the structural model.** This study employed bootstrapping technique with 5,000 bootstraps [109] and 423 cases to generate path values and their significance level as proposed by [113]. Figs 2 and 3 present the assessment of the structural model. [114], proposed that the structural model must evaluate the linear regression effects of used constructs. An

**Table 2. Measurement model.**

| Construct | Item code | Loading | CA | CR | AVE | Inner VIF |
|---|---|---|---|---|---|---|
| Entrepreneurial orientation (EO) | EO1 | 0.796 | 0.923 | 0.937 | 0.649 | 1.611 |
| | EO2 | 0.821 | | | | |
| | EO3 | 0.836 | | | | |
| | EO4 | 0.793 | | | | |
| | EO5 | 0.787 | | | | |
| | EO6 | 0.772 | | | | |
| | EO7 | 0.847 | | | | |
| | EO8 | 0.79 | | | | |
| SM adoption (SM) | | | 0.937 | 0.946 | 0.593 | 1.644 |
| SM for marketing | SMM1 | 0.783 | 0.914 | 0.94 | 0.796 | 2.234 |
| | SMM2 | 0.778 | | | | |
| | SMM3 | 0.794 | | | | |
| | SMM4 | 0.76 | | | | |
| Customer relationship | CR1 | 0.722 | 0.864 | 0.902 | 0.649 | 2.983 |
| | CR3 | 0.773 | | | | |
| | CR4 | 0.798 | | | | |
| | CR5 | 0.732 | | | | |
| | CR6 | 0.722 | | | | |
| Information | IA1 | 0.785 | 0.934 | 0.958 | 0.883 | 2.320 |
| Accessibility | IA2 | 0.795 | | | | |
| | IA3 | 0.792 | | | | |
| Innovation capabilities (IC) | IC1 | 0.787 | 0.865 | 0.902 | 0.648 | 1.058 |
| | IC2 | 0.778 | | | | |
| | IC3 | 0.82 | | | | |
| | IC4 | 0.808 | | | | |
| | IC4 | 0.831 | | | | |
| Firm performance (FP) | FP1 | 0.803 | 0.923 | 0.938 | 0.684 | |
| | FP2 | 0.862 | | | | |
| | FP3 | 0.839 | | | | |
| | FP4 | 0.825 | | | | |
| | FP5 | 0.852 | | | | |
| | FP6 | 0.816 | | | | |
| | FP7 | 0.791 | | | | |

**Table 3. Latent variable correlation and square root of AVE.**

| Construct | 1 | 2 | 3 | 4 | 5 | 6 | 7 |
|---|---|---|---|---|---|---|---|
| Customer relationship (CR) | **0.806** | | | | | | |
| Entrepreneurial orientation (EO) | 0.528 | **0.806** | | | | | |
| Firm performance (FP) | 0.507 | 0.438 | **0.827** | | | | |
| Information accessibility (IA) | 0.714 | 0.583 | 0.457 | **0.94** | | | |
| Innovation capabilities (IC) | 0.17 | 0.021 | 0.183 | 0.196 | **0.805** | | |
| Social media (SM) | 0.631 | 0.609 | 0.525 | 0.841 | 0.18 | **0.77** | |
| Social media for marketing (SMM) | 0.727 | 0.515 | 0.425 | 0.564 | 0.117 | 0.873 | **0.892** |

*Notes*: Values on the diagonal (bold) are square root of the AVE while the off-diagonals are correlations.

**Table 4. Constructs item and descriptive statistics.**

| Construct | Item code | Items | Mean | S.D | Skewness |
|---|---|---|---|---|---|
| Entrepreneurial orientation | EO1 | "Innovations are appreciated above everything else" | 4.035 | 0.941 | −0.823 |
| | EO2 | "We emphasize R&D, technological leadership and innovativeness instead of trusting only those products and services, which we have traditionally found to be good" | 4.09 | 0.926 | −1.04 |
| | EO3 | "We emphasize risk taking" | 3.986 | 0.945 | −0.899 |
| | EO4 | "In our company, many people want to take risk" | 3.934 | 0.94 | −0.877 |
| | EO5 | "Within the last five years, we have brought several new products or services to the market" | 3.917 | 0.894 | −0.713 |
| | EO6 | "We intend to get into markets before our competition" | 3.986 | 0.92 | −0.923 |
| | EO7 | "We are typically ahead of competitors in presenting new products or procedure" | 3.976 | 0.901 | −0.828 |
| | EO8 | "In our company people want to be first in the markets" | 3.995 | 0.867 | −0.733 |
| *SM adoption* | | | | | |
| SM for marketing | SMM1 | "It helps to conduct marketing research" | 4.014 | 0.778 | −1.023 |
| | SMM2 | "It helps to get referrals (word of mouth via likes, shares and followers in Facebook)" | 4 | 0.819 | −0.802 |
| | SMM3 | "It helps to advertise and promote product/services" | 4.064 | 0.786 | −0.788 |
| | SMM4 | "It provides aids to deliver customer services" | 3.988 | 0.821 | −0.776 |
| Customer relationship | CR1 | "It helps to develop customer relations" | 4.163 | 0.853 | −1.099 |
| | CR2 | "Communicate with customers" | 3.87 | 0.822 | −0.78 |
| | CR3 | Conduct customer service activities" | 3.976 | 0.864 | −0.837 |
| | CR4 | "Receive customer feedback on existing product/services | 3.979 | 0.865 | −0.969 |
| | CR5 | "Receive customer feedback on new/future product/services" | 3.986 | 0.864 | −0.767 |
| | CR6 | "Reach new customers" | 4.061 | 0.868 | −1.035 |
| Information accessibility | IA1 | "It helps to search for general information" | 4.277 | 0.84 | −1.015 |
| | IA2 | "Search for competitor information" | 4.236 | 0.892 | −1.045 |
| | IA3 | "Search for customer information" | 4.121 | 0.863 | −1.033 |
| Innovation capabilities | IC1 | "There is constant generation of new product or service ideas in this firm" | 4.035 | 0.9 | −1.025 |
| | IC2 | "We are constantly having R&D funds in order to search for new ways of doing things" | 3.943 | 0.992 | −0.832 |
| | IC3 | "There is creativity in our methods of operation" | 4.071 | 0.886 | −0.651 |
| | IC4 | "This firm is usually a pioneer in the market" | 4.083 | 0.953 | −0.972 |
| | IC5 | "This firm is able to introduce new product or services every five years due to continuous support provided to R&D" | 3.863 | 0.817 | −0.422 |
| SME performance | FP1 | "Improved customer relationship" | 3.943 | 0.767 | −0.977 |
| | FP2 | "Service quality" | 4.007 | 0.939 | −0.925 |
| | FP3 | "Customer engagement" | 3.896 | 0.85 | −0.913 |
| | FP4 | "Increase in company/brand visibility and reputation" | 3.927 | 0.879 | −0.779 |
| | FP5 | "Increased customer loyalty and retention" | 3.993 | 0.838 | −0.834 |
| | FP6 | "Enhance the customer service" | 3.995 | 0.837 | −0.891 |
| | FP7 | "Increase product/service awareness among customers and increase market share" | 4.028 | 0.847 | −0.943 |

evaluation of structural model assessed using path-co-efficient, *t*-value, *p*-value, and coefficients of determination ($R^2$) [102]. According to [115], recommended that $R^2$ values of 0.19, 0.33, and 0.60 are considered weak, moderate, and substantial. The $R^2$ value of 0.371 indicates that 37.1% of SM adoption changes occurred due to EO, while 32.4% of firm performance occurred due to EO, SM adoption, and IC (Table 5). All hypotheses of the proposed model were supported (Table 6).

Aside from the above, this study also used cross-validated-redundancy-measure or effect sizes ($f^2$) and ($q^2$) to assess the proposed model and validate results [116]. [115] also suggested

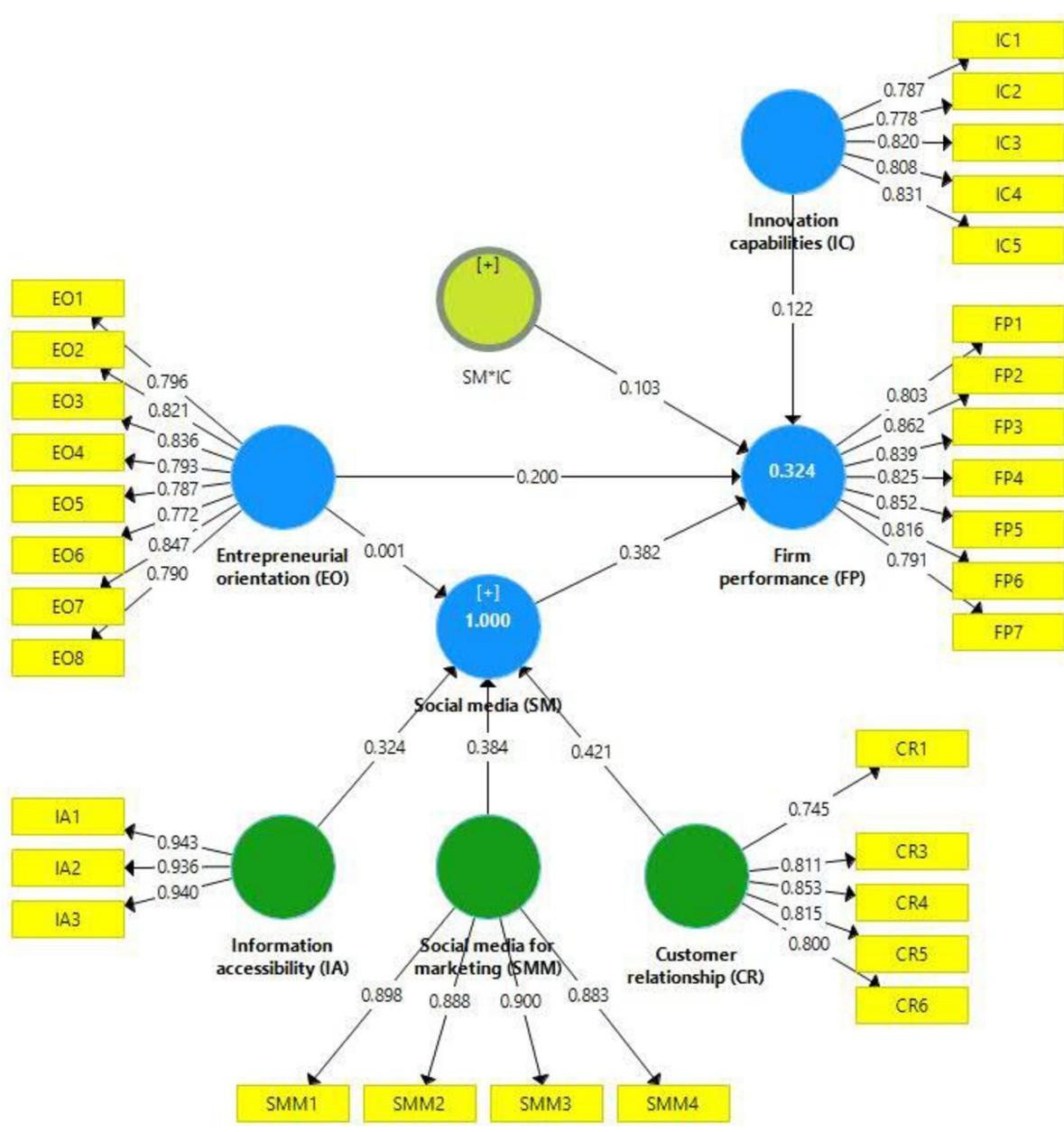

**Fig 2. Structural equation modeling.**

that 0.35, 0.15, and 0.02 values reveal that an exogenous variable has a large, medium, and small predictive relevance for a particular endogenous variable. Predictive relevance is considered a supplementary evaluation due to the goodness of fit index not being appropriate for model authentication as it cannot differentiate between invalid and valid models [117,118]. However, [113] recommended that $q^2>0$ demonstrate that the proposed framework has predictive relevance. This study evidenced that the proposed model has medium and considerable predictive relevance (Table 5). Regarding the moderation results, the constructs had small effects, on an individual basis, and the moderation effect was small (with 0.022 effect size value) (Table 5). In addition, SRMR is widely used as an absolute measure of fit: a value of zero

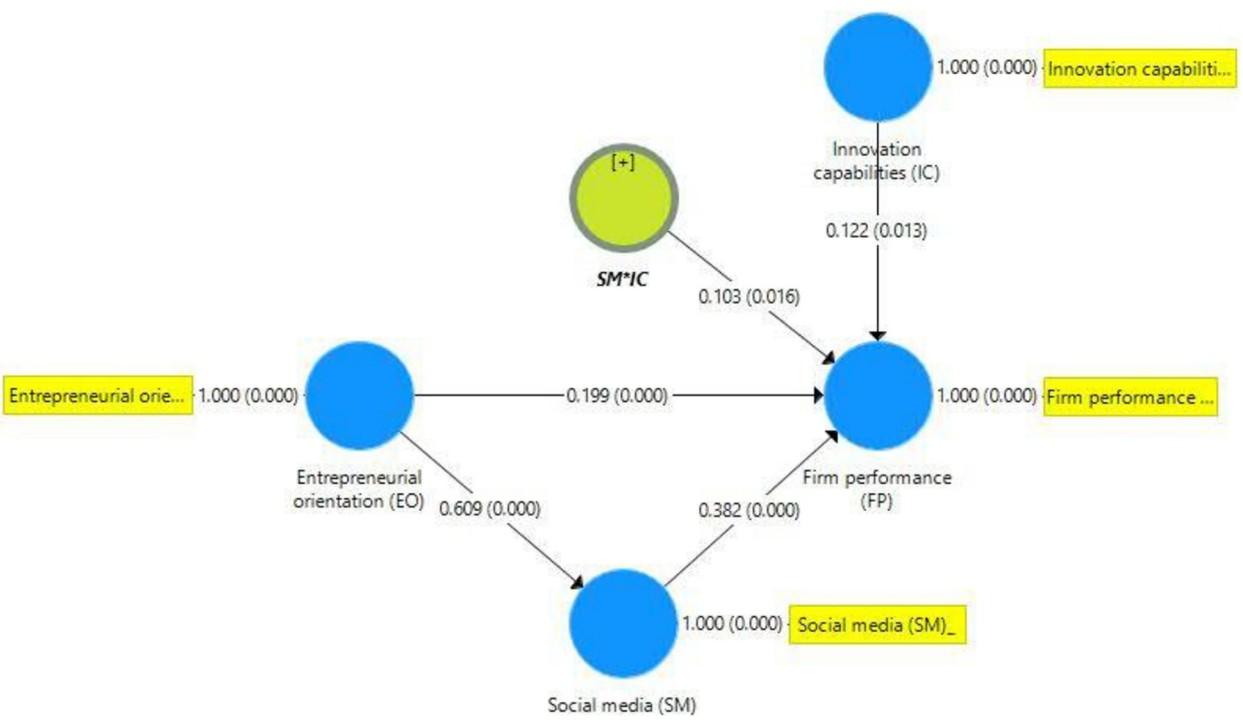

**Fig 3. Structural equation modeling and path coefficients.**

indicates perfect fit and a value less than 0.08 is considered a good fit [119]. Table 5 illustrates that the present research has adequate goodness of fit [120].

**4.2.4 Mediation analysis.** To assess the mediating role of SM adoption,' this study employed [108] method. Following [121], rule $Z$ mediates the link between $X$ and $Y$ if the direct path between $X$ to $Z$ and $Z$ to $Y$ is significant. The results reveal that the direct path from the EO to SM adoption ($\beta = 0.609$, $p = 0.001$) and from SM adoption to firm performance ($\beta = 0.382$, $p = 0.001$) were positive and statistically significant. Accordingly [98], proposed that if

**Table 5. Model strength.**

| Construct | Effect size | | | Coefficient of determination | |
|---|---|---|---|---|---|
| | SSO | SSE | $Q^2$ (= 1–SSE/SSO) | $R^2$ | Adj. $R^2$ |
| SM adoption | 423.00 | 270.663 | 0.360 | 0.371 | 0.369 |
| Firm performance | 423.00 | 295.102 | 0.302 | 0.324 | 0.318 |
| | $f^2$ | | $Q^2$ | | |
| | Firm performance (FP) | Social media (SM) | Firm performance (FP) | Social media (SM) | |
| Entrepreneurial orientation (EO) | 0.037 | 0.590 | 0.03438 (small) | | |
| Social media (SM) | 0.130 | | 0.1361 (small) | | |
| SM*IC | 0.022 | | | | |
| Innovation capabilities (IC) | 0.021 | | 0.02579 (small) | | |
| Information accessibility (IA) | | | | 0.021 (small) | |
| Social media for marketing (SMM) | | | | 0.035 (small) | |
| Customer relationship (CR) | | | | 0.032 (small) | |

**Notes**: Goodness of fit→SRMR = 0.077; SM*IC→moderation effect size is small.

**Table 6. Path coefficient and hypotheses testing.**

| Hypothesis | Relationship | Path coefficient | Mean | SD | *t*-value | Decision |
|---|---|---|---|---|---|---|
| Direct effect | | | | | | |
| *H1* | EO→SME performance | 0.199 | 0.197 | 0.053 | 3.729 | Supported |
| *H2* | EO→SM adoption | 0.609 | 0.609 | 0.043 | 14.274 | Supported |
| *H3* | SM adoption→SME performance | 0.382 | 0.382 | 0.058 | 6.542 | Supported |
| *H4* | IC→SME performance | 0.122 | 0.121 | 0.049 | 2.496 | Supported |
| Indirect effect | | | | | | |
| *H5* | EO→SM adoption→SME performance | 0.233 | 0.233 | 0.04 | 5.751 | Supported |
| Moderation effect | | | | | | |
| *H6* | SM*IC→SME performance | 0.103 | 0.102 | 0.042 | 2.415 | Supported |

*Notes*: Critical values.

* *t*-value>1.96 (*p*<0.05).

the indirect impact is significant while the direct impact is not significant, full mediation has occurred; if both direct and indirect effects are substantial, partial mediation has occurred. SM adoption partially mediated the EO–firm-performance relationship (Table 6).

**4.2.5 Testing the moderation effect.** To assess the moderating role of IC, this study used a product-indicator-method (PIM) using PLS-SEM [122]. We used PIM because the suggested moderating construct was continuous [123]. [115] rules were used to assess the moderating effects. Regarding *H6* (IC moderates the SM-adoption–firm-performance relationship), the interaction terms (β = 0.103, *p* = 0.016) were significant (Table 6, Fig 4). Hence, *H6* was supported. The slope for the link between SM adoption and firm performance moderated by IC

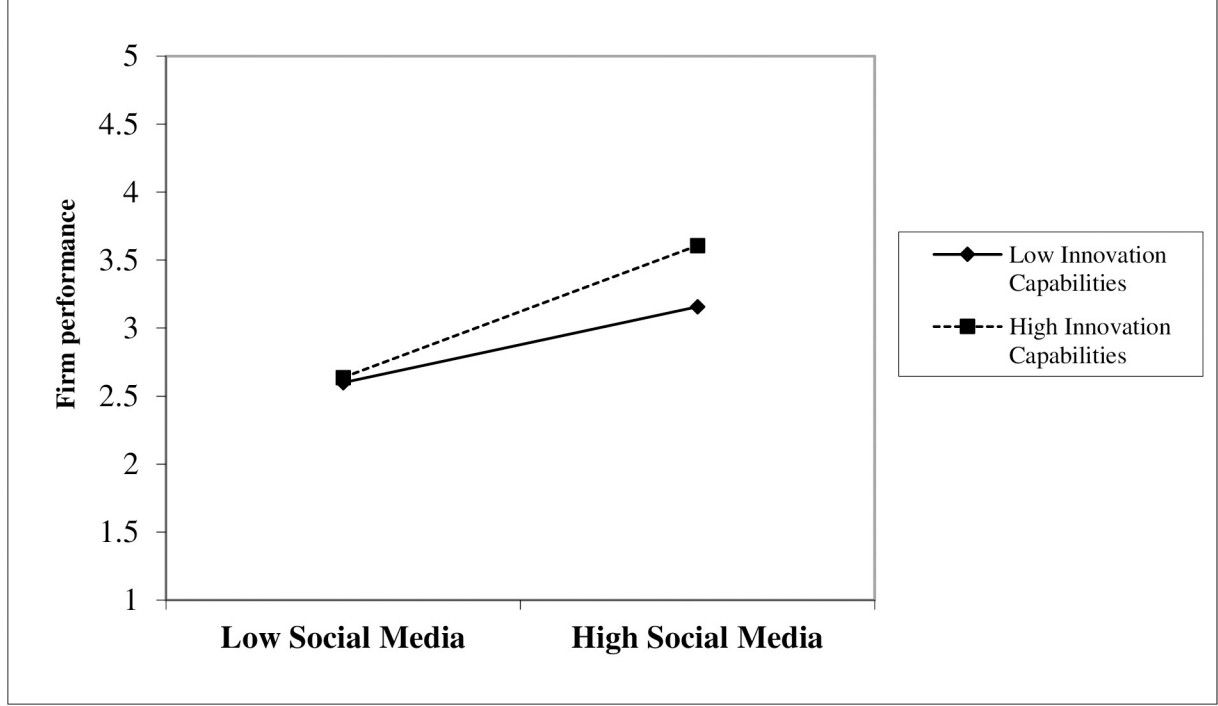

**Fig 4. Moderation effect.**

showed that the relationship became stronger when there was high IC (Fig 4). More specifically, as illustrated in Fig 4, when IC is high, the impact of SM adoption on SMEs performance tends to be stronger.

## 5. Discussion

This study used a quantitative approach to test the EO-SM adoption and SM adoption-SMEs performance relationship. Besides, the mediating role of SM adoption and the moderating role of IC in SMEs context. The study outcomes were found interesting, given that higher EO leads to greater SM adoption; RBV theory supports the assumption that gaining competitive advantage is based on the application of organizations' bundles of productive resources [66].

The EO–SME-performance relationship was positive ($p = 0.001$), supporting *H1*. Results showed that two concepts are relevant, supporting that EO is crucial in enhancing SME performance. To support SME's performance, EO is the essential element that accelerates the organization's SM adoption process. According to RBV, EO is an intangible resource considered the main source of competitive advantage to improve firm performance [40]. Our findings are consistent with [124], who studied SMEs in developing countries, arguing that EO is a significant factor for firm performance.

Regarding the EO–SM-adoption relationship, EO had a significant association with SM adoption ($p = 0.007$), supporting *H2*. These findings are possible due to fear of losing ground to competitors, SMEs in emerging countries like Pakistan are acting entrepreneurially regarding SM adoption. Given increasing SM usage (paradigm and customer shift from offline to online), firms' EO plays a critical role in SM adoption. Our results are consistent with [44] but inconsistent with [18], potentially due to organizations' characteristics, employees', managers', and owners' education, and the risk and costs incurred to encourage staff to adopt SM.

Regarding the effects of SM adoption on SME performance ($p = 0.001$), *H3* was supported. Results showed the robust effect of SM adoption, suggesting that SMEs in emerging countries benefit from investment in SM. This finding is inconsistent with [68], who noted no significant SM-adoption–business-performance relationship among UAE SMEs, but consistent with [13,57], who found the positive effects of SM on firm performance in terms of improved customer relations, cost reduction, and improved information accessibility. These study findings are beneficial for managers, especially in Pakistan, because most of the SMEs do not consider SM as their strategic partner. Still, this study will help them to involve SM to increase firm performance. In this way, they can achieve a competitive advantage in the market. These findings are significant because SM usage by Pakistani firms is uncommon. Firms mainly use SM to market products or services rather than to gain a competitive advantage. SM can enable firms to reduce marketing costs and improve access to information and customer relationships.

IC positively affected firm performance ($p = 0.001$), supporting *H4*. This reflects that IC produces new ideas related to products/services, processes, and marketing activities that improve firm performance. This means that firms will be more proactive in changing trends and will always come up with new and innovative ideas for products and services. The results support [125,126] who found a positive IC–firm-performance relationship. DC theory presumes that competitive advantage enables firms to enhance their capacity to keep up with, respond to, and initiate technological changes. Thus, IC relates not only to firms' propensity to adopt ideas but also a willingness to forgo old habits and engage in the experimental execution of untested ideas. Our findings reveal that Pakistani firms are highly motivated to be innovative and develop IC.

SM adoption partially mediated the EO–SME performance relationship ($p = 0.001$), supporting *H5*. This is consistent with [56], who argued that SM could be adopted to generate

innovative ideas, leading to creative performance and greater effectiveness. The result also found consistent with recent studies of [78,79] who proved the mediating role of SM adoption in the SMEs in developing countries.

Regarding the moderation effect of IC ($p$ = 0.001); *H6* was supported. Our findings suggest that, if SMEs have low IC, SM will have little influence on performance, and vice versa. To summarize, the effects of SM adoption on firm performance are greater when SMEs engage in IC. When a firm is more proactive towards innovation, then they more likely to adopt the SM quickly and increase the firm performance. Because SM adoption has a positive relationship with the SME's performance, and IC plays the role of moderator in their relationship. This is consistent with [70], who found positive moderating effects of IC. Our findings also reflect that IC not only directly influences SME performance but also plays a moderator role in the SM-adoption–SME-performance relationship.

## 6. Theoretical and practical contributions

### 6.1 Theoretical contributions

The present study offers several contributions concerning SM adoption and SMEs performance in emerging economies (an under-researched area). Because in this technological era, customers are more involved in the SM. They prefer SM for shopping, which had made it easier for SMEs to reach potential customers. For SMEs, it is proved as an ample opportunity because SMEs can reach customers through SM. It depends on the capacity of the SMEs innovation level, how much the organization is innovation-friendly and supports innovation in the firm.

We followed the RBV theory, which supports using resources to get a competitive advantage in the market. This theory is supporting this study because we are exploring how firms use their technological resources to get a competitive advantage in the industry. This theory allows us to determine the effect of SM on SMEs' performance. Practically this study will help the manager to explore the relationship between the SM and SMEs performance, and they can decide which organizations can easily go for SM adoption. Because this study will enable the manager to evaluate the chances of SM adoption based on how much the firm environment is innovation-friendly. Because the innovation capability of a firm defines the chances of SM adoption in SMEs. Further managers will be able to demonstrate and decide what to do to increase the performance of the firm through SM.

Although the direct EO–SM adoption and firm performance have been examined in prior studies [38,44], the mediating role of SM adoption [127] and the moderating role of IC has seldom been investigated [128]. This gap is substantial because future innovation adoption may become gradually parallel, with firms hoping to adopt distinctive SM channels for similar purposes. Furthermore, very limited studies have paid attention to SM adoption in SME's context [129,130] and examined hypotheses similar to those in the present study.

This research lengthens the present theory regarding the mediating and moderating role of SM adoption and IC to RBV, then investigating the growing research problem of SM adoption and use by SMEs. The present study offers the newest support to the existing literature by providing empirical aids from an SM viewpoint and its mediating role and moderating role of IC with good explanatory power.

First, we contribute to the body knowledge by analyzing the mediation effect of SM on the EO–SME-performance relationship. Second, this preliminary research enriches the management literature by investigating SM adoption within the moderating confines of IC in SME settings. Theoretically, we investigate the moderator that influences the effects of SM on SME

performance. IC helps to explain the effects of SM adoption in enhancing SME performance. The moderating effects of IC strengthen the SM-adoption–SME-performance relationship. Previous studies have either examined the constructs individually or with different constructs and dimention of EO [131] or called for further research concerning theory development and validation of results [11]. The present study, however, delivered a suitable and parsimonious approach for examining this research phenomenon.

This research guides more in-depth examination and improves research on EO and SM-adoption perceptions by connecting them to firm performance. Additionally, IC has been conceptualized in an innovative-technology and SME context. We confirm the partially mediating role of SM adoption and firm performance in SMEs in emerging countries.

## 6.2 Practical contributions

The present study provides a comprehensive understanding of EO, enabling decision-makers (owners and managers) to recognize the actual consequence of SM adoption. It also supports their understanding regarding how effective SM adoption and implementation can lead to improving SME performance. For instance, SM adoption significantly affects SMEs in terms of cost reduction, innovativeness, competitive advantage related to marketing activities, processes, and work. Actively employing SM also improves firm-customer relationships and customer loyalty. Besides, SM adoption improves brand visibility, enables a significant number of customers to be reached, and enhances customer access to information.

The most widely utilized SM applications were social networking services. Before deciding which application(s) to adopt, decision-makers must work closely with the most popular SM tools in their country [132], i.e. Facebook (Pakistan), WhatsApp (India), and WeChat (China). Firms should encourage employees to use SM for daily work and marketing-related activities as it is a cheaper and faster way to communicate and share information and to innovate. SMEs operating in developing countries should adopt SM as it offers several new ways to do business, thus developing and sustaining IC [125].

We examined EO's effects on SM adoption in developing countries. Future scholars can examine the effects of SM adoption grounded on reflective-formative constructs. Given the debate on the positive (cost-effective, improving customer relationships, brand-building) and negative (increasing customer power, tracking negative comments, marketing-shift burden) SM outcomes, most firms are confused regarding SM adoption. This study clarifies the importance of SM and its outcomes. The findings help firms, especially SMEs, in developing countries. EO helps managers, executives, and employees in SM adoption.

In developing countries, especially Pakistan, the most widely used SM applications were Facebook. Before deciding on which application(s) to adopt and employ for marketing and reaching a large audience, decision-makers must work closely with the most popular SM tools in their country [132], i.e., Facebook (Pakistan), WhatsApp (India), and WeChat (China). Firms should encourage employees to use SM for daily work and marketing-related activities as it is a cheaper and faster way to communicate and share information and to innovate. SMEs operating in developing countries should adopt SM as it offers several new ways to do business, thus developing and sustaining IC [125].

Some scholarships dedicated to SMEs have suggested that firms use SM simply because others in the industry do, resulting in wasted resources and SM adoption, not generating the desired outcomes. We propose that SMEs need to understand how and why SM will be utilized [133]. Ultimately, we highlight why SM should be adopted and how it can be implemented successfully.

## 7. Limitation and future research

This paper is not free from limitations. We used a random sampling method due to time constraints and limited budget. As we considered Pakistan only, limiting the generalizability of the findings, future studies should employ a cross-cultural approach to explore the effects of SM adoption on SME performance and the mediating role of SM adoption between EO and SMEs performance and moderating role of SM adoption and SMEs performance.

There may also be other antecedents and influencing factors; using a single factor and associated elements (e.g., information accessibility, SM for marketing, and customer relationship in SM adoption) may be considered a limitation. The sample size and reach may also be considered a limitation, as it can concentrate on only a few SM tools. This study proposed framework was tested in the context of SMEs in a single country via web-based survey. The results represent a snapshot at a particular time; however the impact of SM is volatile over time. Hypotheses were developed using SMEs operating in developing countries; therefore, the results may not be generalizable to developed countries and larger firms. This study did not examines the effects of the environment (e.g., environmental turbulence, institutional pressure, etc.). Therefore, upcoming studies should examines the effects of environmental turbulence and regulations related to SM marketing content in developing countries.

Future studies could also investigate SMEs in separate sectors, as well as the large organization in emerging economies, needs to be tested. They could also study how managers and executives use SM to manage customers' knowledge. Comparative studies could validate our results and investigate the proposed model using large and small firms. The association between SM adoption and its effects on SMEs' performance in emerging economies could also be studied to investigate whether there is any deviation in results between different timeframes. Finally, the mediating role of SM adoption and the moderating role of IC should be explored further irrespective of country structure.

## Supporting information

**S1 File. Questionare.**
(DOCX)

**S1 Data.**
(CSV)

## Author Contributions

**Conceptualization:** Sikandar Ali Qalati, Muhammad Aamir Shafique Khan, Syed Mir Muhammad Shah, Raza Saleem Khan.

**Data curation:** Sikandar Ali Qalati, Muhammad Aamir Shafique Khan, Syed Mir Muhammad Shah, Raza Saleem Khan.

**Formal analysis:** Mingyue Fan, Sikandar Ali Qalati, Muhammad Aamir Shafique Khan, Syed Mir Muhammad Shah, Muhammad Ramzan, Raza Saleem Khan.

**Investigation:** Sikandar Ali Qalati, Muhammad Aamir Shafique Khan.

**Methodology:** Mingyue Fan, Sikandar Ali Qalati, Muhammad Aamir Shafique Khan, Muhammad Ramzan, Raza Saleem Khan.

**Project administration:** Muhammad Aamir Shafique Khan.

**Software:** Sikandar Ali Qalati, Muhammad Aamir Shafique Khan, Muhammad Ramzan.

**Supervision:** Mingyue Fan, Sikandar Ali Qalati, Syed Mir Muhammad Shah.

**Validation:** Raza Saleem Khan.

**Writing – original draft:** Sikandar Ali Qalati, Muhammad Aamir Shafique Khan.

**Writing – review & editing:** Mingyue Fan, Syed Mir Muhammad Shah, Muhammad Ramzan, Raza Saleem Khan.

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
