## [Decision Letter · Decision Letter 0]

11 Sep 2020

PONE-D-20-22933

Effects of entrepreneurial orientation on social media adoption and SME performance: the moderating role of innovation capabilities

PLOS ONE

Dear Dr. Muhammad Aamir Shafique Khan,

Thank you for submitting your manuscript to PLOS ONE. After careful consideration, we feel that it has merit but does not fully meet PLOS ONE’s publication criteria as it currently stands. Therefore, we invite you to submit a revised version of the manuscript that addresses the points raised during the review process.

We look forward to receiving your revised manuscript.

Kind regards,

Wonjoon Kim, Ph.D

Academic Editor

PLOS ONE

Additional Editor Comments:

Manuscript ID PONE-D-20-22933 entitled "Effects of entrepreneurial orientation on social media adoption and SME performance: the moderating role of innovation capabilities" which you submitted to Tourism Review, has been reviewed. The comments of the reviewer(s) are included at the bottom of this letter.

The reviewer(s) suggest a number of major revisions to your manuscript. Therefore, I invite you to respond to the reviewer(s)' comments and revise your manuscript. Please be aware that the decision with regard to the publication of your paper is depending on the quality of your revisions.

Journal Requirements:

Reviewers' comments:

Reviewer's Responses to Questions

**Comments to the Author**

1. Is the manuscript technically sound, and do the data support the conclusions?

Reviewer #1: Yes

Reviewer #2: No

2. Has the statistical analysis been performed appropriately and rigorously? 

Reviewer #1: Yes

Reviewer #2: No

3. Have the authors made all data underlying the findings in their manuscript fully available?

Reviewer #1: No

Reviewer #2: No

4. Is the manuscript presented in an intelligible fashion and written in standard English?

Reviewer #1: Yes

Reviewer #2: No

5. Review Comments to the Author

Reviewer #1: Thank you for the opportunity to review your manuscript titled « Effects of entrepreneurial orientation on social media adoption and SME performance: the moderating role of innovation capabilities ». The manuscript uses a questionnaire administered to more than 400 Owners/managers/executives in Pakistani SMEs to investigate the effects of entrepreneurial orientation on social media adoption and SME performance in developing countries.

All in all, I believe your work has really good potential to contribute interesting insights that are relevant for the field of organizational studies. However, I feel that this potential remains somehow undeveloped due to the articulation of the contribution and some methodological considerations. I hope to offer some suggestions to revise the manuscript in order to realize its full potential.

So I believe the realizing the potential of your study will call for expanding your analysis. My primary reservation with your manuscript lies with the ultimate message you leave us with, and with the feeling that the paper could go deeper in contributing to the field than it currently does. I would therefore make the following practical recommendations.

First, I would encourage to offer a contribution that is more analytical rather than descriptive of the phenomenon you study, and to make it explicit for the reader from the start : why is SM adoption and IC mediating roles important to understand ? Why should we care ? You build on existing research on the topic, but is it enough to justify more work on the subject ? I feel that this issue needs to be tackle from the introduction.

Second, I have issues with the “Theoretical contributions” (p25). You state that the role of SM adoption and IC are key, especially as “forthcoming technologies may become increasingly similar, with organizations wishing to adopt distinctive platforms for similar purposes ». I don’t see any justification for that statement, which therefore make the contribution kind of flat.

Third, I feel like the “Contribution” section is a mere summary of the findings, and clarifies how it extends (or not) existing litterature. I feel this is not enough. Unpacking a discussion calls for developing the implications of what you found for future research, theorizing, and practice. What would be the ‘big idea’ you’d want me to retain? Would it be only about the fact that SM adoption mediates the effect of EO on performance ? Or might there be something else that is somewhat more general, more conceptual, more abstract yet more generalizable? Would you be able to articulate what these notions add or change to prior knowledge on the use of SM in organizations ? What’s fundamentally new and insightful here? And why is this important? What research ‘problem’ might this help resolve – or what area does this open up? Wouldn’t you want us to discuss these things ? And if so, why not put this in the paper? :-) I’m sure you already have a few ideas. I would strongly recommend that you find ways to better relate your observations to central issues of either EO or SM adoption.

Last, I would expect a discussion regarding the specificities of your study and your results in emerging countries : how/why are things different ? how general are therefore the findings ? For instance, you state that « SMEs’ owners/managers in developing countries are seeking to improve marketing practices via SM adoption, which provides multiple tools to improve firm performance » : how is this specific to developing countries ?

Although I have some concerns with some aspects of your work, I sincerely hope you will have found the few comments and suggestions above helpful and constructive, and I wish you all the very best in your efforts to work with your manuscript.

Reviewer #2: Reviewer report for Plos One jounal

Manuscript ID: PONE-D-20-22933

Manuscript Title: Effects of entrepreneurial orientation on social media adoption and SME performance:

the moderating role of innovation capabilities

All what i need to see is the paper needs more effort

please see my comments

6. PLOS authors have the option to publish the peer review history of their article (what does this mean?). If published, this will include your full peer review and any attached files.

Reviewer #1: No

Reviewer #2: No

---

## [Author Response · Author response to Decision Letter 0]

14 Dec 2020

First we would like to thank the editor and reviewers who gave their efforts and time to suggest and help us in improving overall quality of the manuscript. We have responded to the comments given by reviewers as well as by the editor.

A separate response letter for reviewers has been uploaded.

Regarding the points raised by the editorial staff, we have provided the questionnaire of the study as well as have uploaded our data as well.

You may please update Data Availability Statement as follows:

"All relevant data are within the manuscript and its Supporting Information files.

---

## [Decision Letter · Decision Letter 1]

5 Feb 2021

Effects of entrepreneurial orientation on social media adoption and SME performance: the moderating role of innovation capabilities

PONE-D-20-22933R1

Dear Dr. Muhammad Aamir Shafique Kahn,

We’re pleased to inform you that your manuscript has been judged scientifically suitable for publication and will be formally accepted for publication once it meets all outstanding technical requirements.

Kind regards,

Wonjoon Kim, Ph.D

Academic Editor

PLOS ONE

**Comments to the Author**

1. If the authors have adequately addressed your comments raised in a previous round of review and you feel that this manuscript is now acceptable for publication, you may indicate that here to bypass the “Comments to the Author” section, enter your conflict of interest statement in the “Confidential to Editor” section, and submit your "Accept" recommendation.

Reviewer #1: All comments have been addressed

2. Is the manuscript technically sound, and do the data support the conclusions?

Reviewer #1: Yes

3. Has the statistical analysis been performed appropriately and rigorously? 

Reviewer #1: Yes

4. Have the authors made all data underlying the findings in their manuscript fully available?

Reviewer #1: No

5. Is the manuscript presented in an intelligible fashion and written in standard English?

Reviewer #1: Yes

6. Review Comments to the Author

Reviewer #1: (No Response)

7. PLOS authors have the option to publish the peer review history of their article (what does this mean?). If published, this will include your full peer review and any attached files.

Reviewer #1: **Yes: **Marine Agogué

---

## [Editor Report · Acceptance letter]

18 Feb 2021

PONE-D-20-22933R1 

Effects of entrepreneurial orientation on social media adoption and SME performance: the moderating role of innovation capabilities 

Dear Dr. Khan:

I'm pleased to inform you that your manuscript has been deemed suitable for publication in PLOS ONE. Congratulations! Your manuscript is now with our production department. 

Kind regards, 

on behalf of

Dr. Wonjoon Kim 

Academic Editor

PLOS ONE